# Proline Metabolism Genes in Transgenic Plants: Meta-Analysis under Drought and Salt Stress

**DOI:** 10.3390/plants13141913

**Published:** 2024-07-11

**Authors:** Marco Renzetti, Elisa Bertolini, Maurizio Trovato

**Affiliations:** 1Department of Biology and Biotechnologies, Sapienza University, 00185 Rome, Italy; renzetti.1818304@studenti.uniroma1.it; 2Biocomputing Group, Department of Pharmacy and Biotechnology, Bologna University, 40126 Bologna, Italy; elisa.bertolini5@unibo.it

**Keywords:** proline metabolism genes, drought tolerance, salinity tolerance, transgenic plants, meta-analysis

## Abstract

The amino acid proline accumulates in plants during abiotic stresses such as drought and salinity and is considered a reliable marker of environmental stress. While its accumulation is well established, its precise role in stress tolerance and its underlying molecular mechanism remain less clear. To address these issues, we performed a meta-analysis—a robust statistical technique that synthesizes results from multiple independent studies while accounting for experimental differences. We focused on 16 physiological and morphological parameters affected by drought and salt stress in transgenic plants expressing proline metabolic genes. For each parameter, we calculated the effect size as the response ratio ($\frac{R}{R}$), which represents the logarithm of the mean value in the transgenic group over the mean value of the control group ($\ln\frac{R}{R}$). Under stress, most parameters exhibited significantly higher response ratios in the transgenic group, confirming the beneficial effects of proline during drought and salt stress. Surprisingly, under non-stressed conditions, most stress markers showed no significant differences between transgenic and non-transgenic plants, despite elevated proline levels in the former. These results suggest that the benefits of proline may be related to proline catabolism or may only become apparent during stress, possibly due to interactions with reactive oxygen species (ROS), which accumulate predominantly under stress conditions.

## 1. Introduction

The rapid rise in the global population, combined with the challenges brought about by global warming, creates significant pressure on agriculture to satisfy the increasing food demand while also ensuring environmental sustainability. To tackle this issue, it is essential to investigate new strategies in agriculture [1,2], such as developing drought- or salt-resistant crops, and to comprehend plant mechanisms aimed at securing high crop yields in unfavorable environmental conditions. Because of global warming and its associated rapid and extreme climatic changes, the development of drought- or salt-tolerant crops has become a major objective of plant scientists worldwide. Proline accumulation upon environmental stress, especially drought and salt stress, is one of the most widespread responses among plant species, suggesting that proline may contribute to drought and salinity tolerance [3,4]. Accordingly, several transgenic plants for proline metabolism genes have been generated over the last decades to increase plant tolerance to abiotic stresses by increasing their proline levels. In higher plants, proline metabolism uses short pathways that are easily modified by genetic engineering. Proline synthesis takes place in the cytosol and can start from either glutamate or ornithine, which are reduced to $\delta^{1}$-pyrroline-5-carboxylate (P5C) by 1-pyrroline-5-carboxylate synthetase (P5CS) and ornithine aminotransferase (OAT), respectively. P5C spontaneously cyclizes to glutamate-5- semialdehyde (GSA), which is further reduced to proline by $\delta^{1}$-pyrroline-5-carboxylate reductase (P5CR). Proline catabolism occurs in the mitocondrium, where proline is oxidized back to GSA and glutamate by the sequential action of proline dehydrogenase (ProDH) and $\delta^{1}$-pyrroline-5-carboxylate dehydrogenase (P5CDH) [5,6]. Several studies have shown that transgenic plants engineered with proline synthesis genes exhibit increased proline production, performing better than non-transgenic controls under a range of different environmental stresses [3]. However, while most of the studies share the conclusion that proline is a reliable marker of stress, its role in stress tolerance, and the underlying mechanisms of proline’s protective effects, remain controversial. Some authors argue that proline’s accumulation may simply be a byproduct of the plant’s stress response mechanisms, rather than an active contributor to tolerance [7,8,9]. Another controversial topic about proline biology is centered around the mechanism responsible for stress tolerance. Due to its properties as a compatible osmolyte, capable of accumulating in cells without damaging cellular metabolism, the protective role of proline against water stress was thought to depended on the intracellular increase in osmotic pressure. However, while this mechanism can hold in some cases, many other functions attributed to proline, such as kosmotropic agent, redox buffer, ROS scavenger, and signaling molecule [5], could also explain the beneficial effects of proline on water stress, leaving the question of proline’s mechanism of action unresolved. Similarly, the question of whether proline or its metabolism is involved in stress tolerance remains an open problem [5,8,10]. Consistently, despite a large body of literature claiming the beneficial effects of proline on stress tolerance, the protective role of proline against stress remains a topic of debate, as does the specific mechanism responsible among those proposed in the literature. It is precisely from the analysis of the many works on proline, sometimes with conflicting results, that we have started this work, trying to answer these questions with a meta-analytic approach to integrate results from multiple studies and quantitatively evaluate the effect of proline on drought and salinity tolerance in plants. Through this robust statistical analysis, we measured the effects of proline overexpression on 16 stress-related physiological and morphological parameters to clarify the role of proline in enhancing stress tolerance. In particular, we addressed the following questions:Does proline accumulation under drought and salt stress confer stress tolerance to plants’ transgenic for proline metabolism genes, or is it merely a consequence of plant stress?What is the role of osmotic adjustment in improving stress tolerance in plants transgenic for proline metabolism genes?Is it proline metabolism or proline itself that confers tolerance to plants under conditions of drought and salt stress?Can proline increase tolerance to drought and salinity by acting as an ROS scavenger?

The results of this meta-analysis may help clarify the role of proline in stress tolerance and improve our understanding of its underlying molecular mechanisms, helping plant breeders and molecular biologists to design optimal strategies to develop stress-resistant crop varieties.

## 2. Results

### 2.1. Summary Effects

The natural logs of the means of transgenic plants for proline metabolism genes over those of non-transgenic plants ($\ln\frac{Rt}{Rc}$) are shown in the forest plots of Figure 1 and Figure 2 for 16 morphological and physiological factors under stressed (Figure 1) and non-stressed (Figure 2) conditions, respectively. The physiological factors have been chosen because they are affected by drought and salt stress and represent reliable indicators of stress tolerance. Values of $\ln\frac{Rt}{Rc}$ greater than 0 indicate that the transgenic plants for proline metabolism genes have a positive effect on stress tolerance, while negative values indicate an adverse effect. In the case of malondialdehyde activity (MDA) and relative electric conductivity (Rec), however, negative values indicate a positive effect and positive values indicate a negative outcome. Our meta-analysis covers 199 studies from 47 papers (Appendix A) and includes 100 dicots and 99 monocot species belonging to 7 families (*Brassicaceae, Convolvulaceae, Poaceae, Solanaceae, Fabaceae, Rosaceae, and Rutaceae*). Tobacco (*Nicotiana tabacum, and Nicotiana plumbaginifolia*), belonging to the *Solanaceae* family, was the most represented dicot (54 studies), and rice (*Oryza sativa*), belonging to the *Poaceae* family, was the most represented monocot (44 studies). The CaMV35S was the most used promoter (133 studies), and *P5CS* was the most represented donor gene (121 studies). As shown in Figure 1, under stress conditions most of the parameters (13 out of 16) showed a response ratio significantly higher in the transgenic group than in the non-transgenic one, and another parameter, MDA, was significantly lower in transgenic plants. Overall, 14 out of 16 parameters were indicative of a positive effect of proline on drought and salt stress tolerance, and only two effect sizes (SOD and Rec) were not significant. Under non-stressed conditions, in contrast (Figure 2), nearly all the parameters showed no differences between transgenic and non-transgenic plants (13 out of 16), with the notable exception of proline, and root length, which showed significantly increments of 64%, and 11%, respectively, and of catalase activity, which showed a non-significant increment of 15%. The parameters with the most significant effect sizes (*p* < 0.001) under stress conditions were proline content, plant height, seed weight, chlorophyll content, root length, peroxidase activity, malondialdehyde activity, relative electrical conductivity, and plant survival. Among these, the ones with the largest difference between transgenic and non-transgenic plants were plant survival, proline content, root weight, and root length, with values 139%, 94%, 80%, and 56% higher in transgenic than in non-transgenic plants. Four parameters (plant weight, CAT, APX, and RWC ) were significant, with a *p*-value less than 0.1, and one parameter (stomatal aperture) was significant, with *p* < 0.05. For ease of interpretation, in the last column of Table 1 and Table 2 we reported the fold variations under stress (Table 1) and non-stress (Table 2) conditions, respectively, in exponentiated form.

### 2.2. Heterogeneity and Moderation Analysis under Drought and Salinity Stress

The effect sizes for parameters under stress conditions (Table 1) show highly significant Q statistic *p*-values (*p* < 0.001), indicating heterogeneity primarily due to study differences, as evidenced by the $I^{2}$ statistics. Therefore, we performed a moderation analysis to investigate the causes of this heterogeneity and identify the potential moderators capable of influencing the effect sizes. Because of the relatively small number of studies of our meta-analysis, we used the method of Hunter and Schmidt [11] to calculate $\tau^{2}$ and estimate the amount of heterogeneity. This method is often used in meta-analyses with small numbers of studies because it does not assume a specific distribution for the effect sizes and is more sensitive to heterogeneity. The default REML method used by metafor, on the contrary, is based on large sample theory under the assumption that the parameter estimates are asymptotically normally distributed and can be biased by incorrect asymptotic approximations, especially when the sample size is small. Furthermore, to increase the robustness of our analysis, we also performed a permutation test [12]. This test randomly rearranges the rows of the model matrix to evaluate all possible intercepts and compares the resulting *p*-value against the distribution of possible p-values.

### 2.3. Detailed Moderation Effect Analysis

Following our investigation into the heterogeneity of effect sizes under drought and salinity stress, we now present a detailed moderation effect analysis for each of the 16 parameters studied.

#### 2.3.1. Proline

Transgenic plants with proline metabolism genes consistently produced more proline than non-transgenic controls (Figure 2). The type of promoter and the recipient species, however, significantly influenced proline accumulation, explaining 29.31% and 26.26% of the total heterogeneity, respectively. This was confirmed by a significant omnibus test for moderation (*** for promoter and * for recipient species) and permutation tests (Appendix A).

#### 2.3.2. Plant Height

All moderators, except treatment, impacted plant height to some degree, with soil cultivation having the most positive effect (Appendix A). Conversely, seed number was unaffected by any moderator except treatment, which accounted for 85.34% of the total variability (QM *p*-value = 0.0008) (Appendix A).

#### 2.3.3. Seed Weight and Chlorophyll

Seed weight was influenced by all moderators, with treatment being a consistent factor (Appendix A). Chlorophyll content was significantly modulated by the donor gene type ($R^{2}$ = 48.75%; QM *p*-value < 0.0001; permutation test = *) and treatment type ($R^{2}$ = 57.81%; QM *p*-value = 0.0005) (Appendix A).

#### 2.3.4. Root Length and Plant Weight

The severity of treatment (mild, moderate, severe) notably affected root length and plant weight, explaining 70.48% and 67.35% of the total heterogeneity, respectively, with significant QM values (0.0011 for root length, which did not pass permutation testing; <0.0001 for plant weight) (Appendix A).

#### 2.3.5. Peroxidase (POD), Superoxide Dismutase (SOD), Malondialdehyde (MDA), Catalase (CAT), and Ascorbate Peroxidase (APX)

Because of the low number of studies, it is difficult to safely assess if and how different moderators can affect the peroxidase activity of transgenic plants, although the type of treatment, particularly if moderate, seems to have a strong effect on POD activity (Appendix A). Superoxide dismutase activity was affected by the type of donor gene, the type of recipient species, the medium type, and the generation, with the latter exerting the strongest moderation effect, explaining 86.64% of total heterogeneity with a highly significant omnibus test for moderation (<0.0001) confirmed by the permutation test (Appendix A). The activity of malondialdehyde (MDA), a lipid peroxidation marker considered a reliable indicator of oxidative stress, was mainly affected by the donor gene, recipient species, and, to a lesser extent, promoter type (Appendix A). Intriguingly, the type of treatment showed no effect on MDA activity. Catalase activity, on the contrary, was strongly affected by the type of treatment, as well as by generation and recipient type, but not by the medium nor the donor gene (Appendix A). Ascorbate peroxidase (APX) activity appeared mainly moderated by the donor genes, especially by *P5CS*, by the recipient species, particularly belonging to the *Solanaceae* family, and, to some extent, by being in the T1 generation (Appendix A).

#### 2.3.6. Relative Water Content (RWC)

All moderators, except treatment, affected RWC, particularly the donor gene, which explained 67.41% of the total variability (QM *p*-value =< 0.0001, permutation test = **), and the recipient group, which accounted for 67.41% of the total variability (QM *p*-value =< 0.0001, permutation test = **) (Appendix A).

#### 2.3.7. Stomatal Conductance, Relative Electric Conductance, and Survival

No moderators seem to modulate significantly, either stomatal aperture (Sto) or relative electric conductivity (Rec). However, the low number of studies suggests taking these conclusions with care. Survival, on the contrary, might be influenced by all moderators and particularly by treatment, but the low number of studies renders these conclusions less reliable (Appendix A).

### 2.4. Heterogeneity and Moderation Analysis under Non Stress Conditions

Although under non-stressed conditions the majority of stress markers (14 out of 16) showed comparable responses in transgenic and non-transgenic plants, with the notable exceptions of proline and root length, 12 out of 16 moderators displayed significant between-study heterogeneity, justifying a moderation analysis (shown in Appendix A). It is important to note that the results of this analysis should be taken with extreme caution since the number of studies under non-stress conditions is particularly low. As in the case of moderation under stress, the results tend to be moderator and level specific. Overall, generation and, to a lesser extent, medium were the least influential moderators, while promoter and, to a lesser extent, donor gene and recipient species were the most influential moderators (Appendix A).

## 3. Discussion

Proline accumulation is a widespread phenomenon that occurs in most plant species under stress conditions, particularly of environmental types [3,4] (Figure 3).

Accordingly, this amino acid is considered an important marker of stress, although its role in stress tolerance and its mechanism of action are still debated [5,13,14]. Because of its peculiar properties as a compatible osmolyte that can accumulate in plant cells without disturbing cell metabolism, the accumulation of proline during environmental stress was initially interpreted uniquely as a plant strategy to limit water loss by increasing the osmotic pressure and decreasing the cellular water potential (ΨW) in plant cells [5]. However, this strategy is not universally conserved among plant species [13] and may be less effective under extreme conditions. For example, halophytes predominantly utilize inorganic ions for osmotic regulation, conserving energy that would otherwise be expended on synthesizing organic ions like proline, thereby avoiding potential yield penalties associated with energy-intensive processes [1]. Furthermore, the idea that high levels of proline may avoid water withdrawal by lowering cellular ΨW has been challenged by some authors because, in most cases, the concentrations of proline measured in plant cells appeared too low to support an osmotic role for proline [5,7,15]. More recently, beyond its roles in protein synthesis and osmoprotection, additional and multiple functions have been tentatively proposed for proline, including being a chelator of toxic metal ions [14,16], ROS scavenger [17,18], redox buffer [19], kosmotropic molecule [5], and signaling molecule [20,21,22,23], capable of modulating the expression of stress-responsive genes and orchestrating a comprehensive defense strategy. At present, it is not clear whether these multiple functions rely on proline accumulation or on its peculiar metabolism, where synthesis and catabolism occur in distinct subcellular compartments [8,10]. Additionally, several reports have revealed the importance of proline in plant development, where this amino acid has been found to affect physiological processes as important as flower transition [21,24], bolting [25,26], embryo development [27,28], pollen development and fertility [29,30,31], and root development [10,32], suggesting a developmental role that extends beyond its stress-related functions.

To elucidate the complex interplay between proline and plant stress tolerance, we performed a meta-analysis to combine the results of several independent studies focusing on the effects of the transgenic expression of the genes involved in proline metabolism on drought and salt stress tolerance. By analyzing 199 studies across 47 publications with an emphasis on 16 morphological and physiological parameters, we have dissected the impact of genetically modified proline pathways in plants. Although the use of transgenic plants to increase proline accumulation has been largely superseded by less problematic, non-transgenic techniques, such as precision breeding, TILLING, or gene editing [33,34], the results of these works represent a rich source of information that deserves to be meta-analyzed to improve our knowledge on the role of proline in stress resistance, and to unveil its underlying mechanism. Under drought and salt stress conditions, the pronounced increment in response ratios for most parameters in transgenic plants compared to non-transgenic ones consolidates the hypothesis that proline confers a protective advantage in stressful environments. Indeed, almost all the physiological and morphological traits chosen as stress markers, including important agronomic traits such as plant survival, seed weight, and root length, exhibited significant enhancements in transgenic plants, highlighting the potential of exploiting proline metabolism genes to fortify crops against drought and salinity stresses. It is also crucial to explore the proline metabolism under combined stress conditions like drought and salinity [35]. Since under non-stressed conditions, the manipulation of proline metabolism does not detrimentally affect plant growth, these results are particularly encouraging and reinforce the strategy of modifying proline pathways to mitigate the effects of drought and salt stress. It is interesting to note that, in non-stressful conditions, there are no significant differences between transgenic and non-transgenic plants, at least relative to the parameters analyzed in this work, with the exclusion of proline content and root length. Intriguingly, root length and catalase activity have been reported by Baudin et al. (2022) to be modulated by proline metabolism under normal developmental conditions, supporting the role of proline in root development [10]. This indication, however, must be taken with caution because of the small number of studies scrutinized in this meta-analysis in support of this hypothesis. The fact that, despite the high proline levels produced by the transgenic variants, almost all the stress markers exhibited similar values under non-stressed conditions suggests that the protective benefits of proline do not depend on proline synthesis and accumulation within the plant cells but rather on its catabolism. Alternatively, the beneficial effects of proline may manifest primarily under stress conditions because of proline’s capacity to interact with and neutralize reactive oxygen species (ROS), which are known to rise under stressful conditions. This alternative hypothesis is corroborated by the increase (or decrease in the case of MDA) of four ROS scavengers (peroxidase, malondialdehyde, catalase, and ascorbate peroxidase), suggesting a possible interplay between proline accumulation and ROS metabolism, as reported by different authors [17,20,33,36,37]. Surprisingly, the average SOD activity, which catalyzes the quick dismutation of superoxide to hydrogen peroxide, displays similar values in both transgenic and non-transgenic plants, suggesting that proline metabolism specifically affects certain ROS detoxification pathways, but not the superoxide dismutation pathway. We cannot rule out, however, the possibility that compensatory mechanisms are activated to maintain stable SOD levels despite the overexpression of proline. With regard to the putative osmoprotective role of proline, the results of this meta-analysis tend to minimize its importance, in agreement with the reports of Bhaskara et al., 2015; Forlani et al., 2019; Kavi Kishor and Sreenivasulu 2014; Ben Rejeb et al., 2014; Sharma et al., 2011; and Signorelli 2016 [5,7,8,9,36,38]. Although statistically significant, the effect size of RWC shows a modest decrease of 1%, which seems insufficient to prevent dehydration and sustain drought or salt stress tolerance. Regarding the relationship between stress tolerance and productivity, drawing firm conclusions remains challenging. Furthermore, advanced approaches are highly warranted and offer promising results. For instance, a spatiotemporal mapping of leaf apoplastic ion and metabolite patterns is a valuable approach that enables the simultaneous capture of numerous compounds, including ions, metabolites, proteins, hormones, and others, with a single sampling event. Consequently, it advances the study of the leaf apoplast’s roles in cell–cell communication and being a conduit for metabolite trafficking, like proline [39]. However, despite the limitations of this research, the data suggest that plants genetically modified to enhance proline metabolism demonstrate increased productivity under stress conditions. In this context, it is crucial to recognize the inherent limitations associated with meta-analytic methods, especially when conducted with a relatively limited number of studies. To mitigate this concern, we implemented the approach developed by Hunter and Schmidt, which was specifically tailored to improving the analysis of small study datasets. However, cautious interpretation of our results remains essential. Another point to take into account is the high heterogeneity, which, as indicated by the Q statistic and the $I^{2}$ statistic, is largely based on real differences between studies. Although the moderation analysis highlighted the importance of some moderators in specific parameters, it failed to isolate a common pattern, although, based on the rigorous permutation test, medium, generation, and treatment seem to have more influence than donor gene, recipient species, and promoter.

## 4. Conclusions

In conclusion, our data and meta-analysis show that transgenic plants for proline metabolism genes exhibit enhanced tolerance to drought and salinity, with no negative effects under normal conditions, confirming a positive role of proline metabolism under stress conditions.

In transgenic plants for proline metabolism genes, proline accumulates in large amounts under both stress and non-stress conditions. However, under non-stress conditions, proline accumulation has no effect on stress markers, suggesting that the positive effects of proline on stress tolerance rely on proline metabolism rather than on proline itself.

Moreover, despite the known properties of proline as a compatible osmolyte, osmotic regulation might play only a marginal role in stress tolerance, whereas redox regulation and interactions with ROS, which accumulate predominantly under stress conditions, might play a major role.

The overall effects of the ectopic expression of proline metabolism genes on stress tolerance, across all the studies, justify the interest in the genetic modification of the metabolism of this amino acid. However, due to the high heterogeneity and significant effects of different moderators, a tailored, case-by-case approach is recommended for proline-related genetic modifications.

## 5. Materials and Methods

### 5.1. Data Collection

To collect tha data, we scrutinized the scientific literature by searching the electronic databases Scopus, PubMed, and Google Scholar for a combination of terms such as “proline”, “P5CS*”, “*pyrroline-5-carboxylate”, “P5CR”, “ProDH”, “ Ornithine-delta-aminotransferase”, “OAT”, “transgen*”, “drought”, “salt”, “salinity”, and “plants”. To refine the search, we manually analyzed the cited literature to discover relevant documents missed from, or absent in, the electronic databases. The search was carried out up to December 2022. The inclusion/exclusion criteria were as follows: (Figure 4):(a)Only transgenic plants of any plant species, including insertional mutants;(b)Only drought or salinity stress;(c)Only proline metabolic genes (both anabolic and catabolic);(d)No exogenous proline treatments;(e)No mutants or allelic variant;(f)All the measures are expressed as fresh weight.

Overall, we collected information from 199 studies extracted from 47 articles, and the references for these studies can be found in the “references” table of Appendix A. Multiple treatments belonging to single articles were treated as independent observations, as commonly used in plant meta-analysis [40,41,42,43]. Furthermore, we calculated effect sizes, taking into account non-independent observations with the “robust” function of the “metafor” package [44,45,46], and found no significant differences compared to the effect sizes calculated without correction for the non-independence of the studies. Means, sample sizes, and variances were obtained from the original studies when possible, otherwise they were extracted from the article figures using the free, open-source application ImageJ version 1.53a [47].

### 5.2. Effect Size and Moderation Analysis

In this study, a meta-analysis was conducted with the R “metafor” package [48] to synthesize the effects of transgenic plants for proline metabolism genes on 16 physiological and morphological parameters regarded as stress markers. We used the natural logarithm of the response ratio ($\ln\frac{R}{R}$) as an effect size, $\ln\frac{R}{R}\;=\;\ln\frac{Rt}{Rc}$, where Rt is the mean of transgenic plants and Rc is the mean of non-transgenic control plants. The response ratio is frequently used in plant biology because, being a dimensionless measure, it can be used across different studies and with different measure units [49,50,51]. Moreover, the natural logarithm of the response ratio allows for symmetrical distribution and ease of interpretation, with values greater than zero indicating a positive effect, values less than zero indicating a negative effect, and a value of zero indicating no effect [52]. It is important to note that, in the case of malondialdehyde activity (MDA) and relative electrical conductivity (Rec), negative values represent positive outcomes and are indicative of stress tolerance. Because of the high levels of heterogeneity due to among-study variations, we utilized the metafor package to compute the weighted mean effect size across studies using the DerSimonian and Laird random-effects model. All the analyses were conducted using R (version 4.2.3) [48]. To investigate the cause of the among-study heterogeneity and identify possible modulators of the effect sizes, we carried out a moderation analysis. The moderator variables included in the model were donor (trans)genes, promoter types, family of the recipient species, taxonomic group of the recipient species (dicot or monocot), growth medium, transgene generation, and treatment type. Consistent with the aim of the work, we focused on the effect sizes estimated under stress conditions. To reduce bias and errors caused by a limited number of studies, when possible, we grouped studies into larger groups, for example classifying species into families and grouping treatments into three groups according to severity (mild, moderate, and severe).

### 5.3. Meta-Analysis

The meta-analysis was carried out using the response ratio as the outcome measure and fitting the data with a random-effects model. The amount of heterogeneity (i.e., $\tau^{2}$) was estimated using the Hunter–Schmidt estimator [11,53]. In addition to the estimate of $\tau^{2}$, the Q-test for heterogeneity [54], the $I^{2}$ statistic [55], and the prediction intervals are reported in Table 1 and Table 2 for stress and non-stress conditions, respectively. To identify potential outliers, we employed a conservative strategy combining Studentized residuals, Cook’s distances, and visual inspection of forest plots to ensure the robustness of our findings [56]. In the absence of obvious errors, we decided to remove only the most extreme outliers without recalculating the effect sizes on the modified datasets. Studies with a Studentized residual of 2.5 standard deviations away from the predicted value were considered potential outliers and removed from the dataset. Studies with a Cook’s distance larger than the median plus six times the interquartile range of the Cook’s distances are considered to be potentially influential were also removed. Finally, we visually examined the forest plots to make a further assessment of the distribution and confidence intervals of the error sizes. To correct for possible artifacts and biases that can occur when the number of studies is small, we used the Hunter and Schmidt method [11] to estimate $\tau^{2}$. This method uses an adjusted method to calculate the sample variances of the correlation coefficients, the sample sizes as weights in the analysis, and the Hunter and Schmidt estimator to calculate the amount of heterogeneity [11]. The analysis was carried out using R (version 4.2.3) [48] and the metafor package (version 4.2.3) [46]. All R codes and datasets used in this meta-analysis are available in the “R codes” folder in the Appendix A.

## Figures and Tables

**Figure 1 plants-13-01913-f001:**
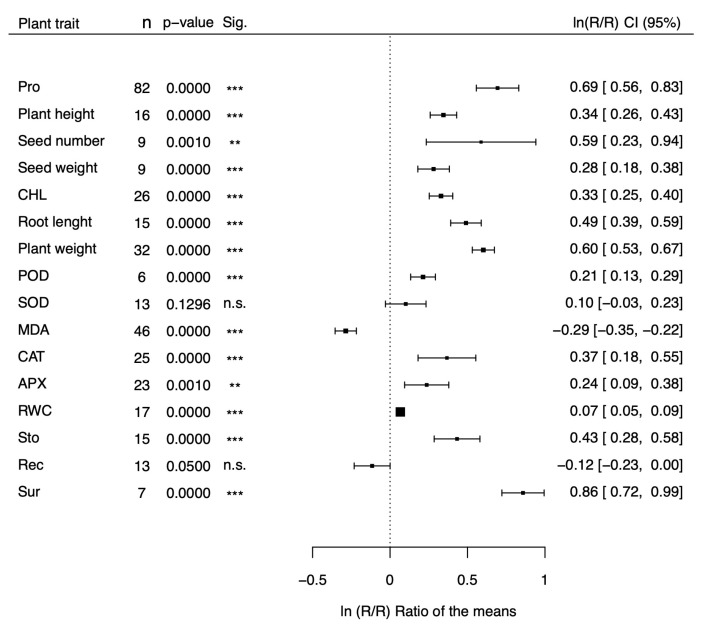
Forest plot showing the effect size ($\ln\frac{R}{R}$) on several stress-related physiological parameteres of transgenic plants for proline synthesis genes compared to non-transgenic plants under drought and salt stress. Pro = Proline content; CHL = Chlorophyll content, POD = Peroxidase activity; SOD = Superoxide dismutase activity; MDA = Malondialdehyde activity; CAT = Catalase activity; APX = Ascorbate peroxidase activity; RWC = Relative water content; Sto = Stomatal conductance: Rec = Relative electric conductivity; Sur = Survival rate. ** *p* < 0.01, *** *p* < 0.001, n.s. = non statistically significant.

**Figure 2 plants-13-01913-f002:**
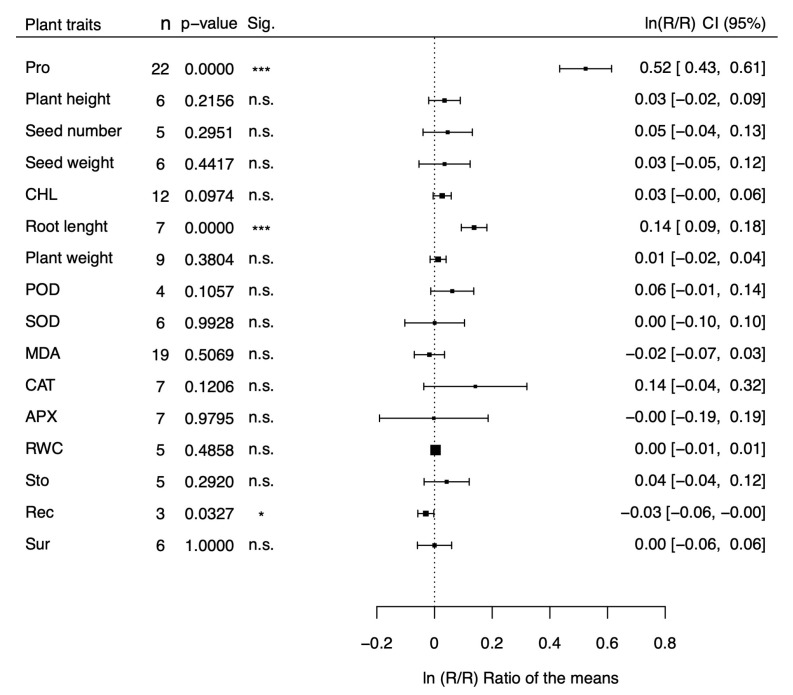
Forest plot showing the effect size ($\ln\frac{R}{R}$) on several stress-related physiological parameteres of transgenic plants for proline synthesis genes compared to non-transgenic plants under non-stressed conditions. Pro = Proline content; CHL = Chlorophyll content, POD = Peroxidase activity; SOD = Superoxide dismutase activity; MDA = Malondialdehyde activity; CAT = Catalase activity; APX = Ascorbate peroxidase activity; RWC = Relative water content; Sto = Stomatal conductance: Rec = Relative electric conductivity; Sur = Survival rate. * *p* < 0.05, *** *p* < 0.001, n.s. = non statistically significant.

**Figure 3 plants-13-01913-f003:**
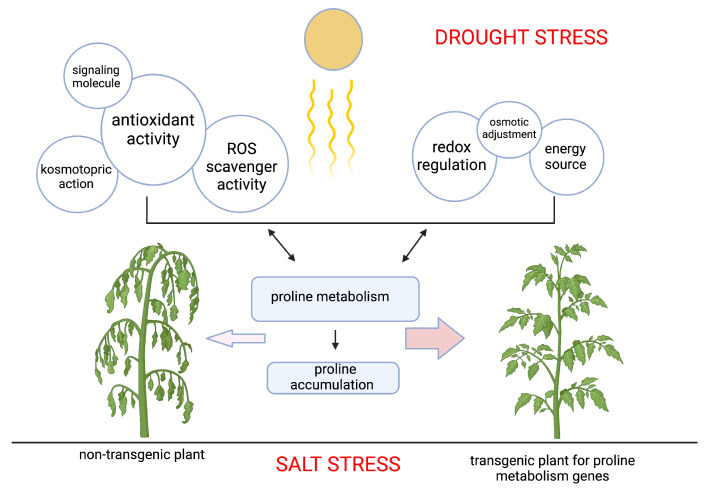
Enhanced tolerance to drought and salt stress in transgenic plants with proline metabolism genes. This figure visualizes the hypothetical roles of proline accumulation and metabolism, with rectangle and bubble sizes scaled according to their relative importance as determined by our meta-analysis findings. Figure created with BioRender.com.

**Figure 4 plants-13-01913-f004:**
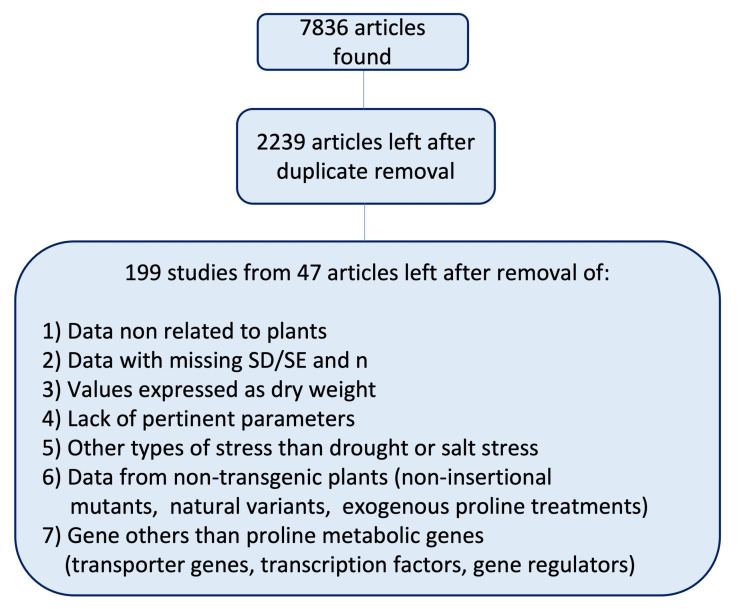
Study selection diagram showing the workflow and the inclusion/exclusion criteria used in this meta-analysis.

**Table 1 plants-13-01913-t001:** Heterogeneity statistics and fold change under drought and salt stress. POD = Peroxidase activity; SOD = Superoxide dismutase activity; CAT = Catalase activity; APX = Ascorbate peroxidase activity; RWC = Relative water content. Q test = *p*-value of the Q statistic (which measures heterogeneity) under the null hypothesis that there is no true heterogeneity among the studies. PI = Prediction interval; $\ln\frac{R}{R}$ = Response Ratio effect size; exp = to make the data more intuitive we have exponentiated the effect size with the formula: $\frac{R}{R}\;=\;e^{\ln\left(\frac{R}{R}\right)}$; % change = percentage fold change obtained with the formula (exp − 1) × 100.

Parameter	$\tau^{2}$	Q Test	$I^{2}$	PI	$\ln\frac{R}{R}$	Exp	% Change
Proline	0.3906	0.0000	99.53%	0.53/1.92	0.6939	2.00	100%
Plant height	0.0234	0.0000	97.63%	0.03/0.65	0.3445	1.44	44%
Seed number	0.2817	0.0000	99.33%	0.60/5.40	0.5874	1.80	80%
Seed weight	0.0155	0.0000	81.21%	1.02/1.73	0.2810	1.32	32%
Chlorophyll	0.0250	0.0000	83.54%	1.01/1.91	0.3289	1.39	39%
Root length	0.0279	0.0000	98.74%	0.15/0.83	0.4400	1.63	63%
Plant weight	0.0264	0.0000	96.37%	1.32/2.53	0.6021	1.83	83%
POD activity	0.0074	0.0000	92.00%	1.03/1.49	0.2129	1.24	24%
SOD activity	0.0550	0.0000	98.43%	0.69/1.78	0.1013	1.11	11%
MDA activity	0.0434	0.0000	92.84%	−0.50/−1.14	−0.2860	0.75	−25%
CAT activity	0.2044	0.0000	96.39%	0.58/3.57	0.3671	1.44	44%
APX activity	0.0909	0.0000	92.36%	−0.37/0.84	0.2366	1.26	26%
RWC	0.0012	0.0000	83.68%	1.00/1.15	0.0665	1.07	7%
Stomatal conductance	0.0461	0.0000	81.12%	0.01/0.87	0.4325	1.54	54%
Electric conductivity	0.0365	0.0000	97.28%	0.94/1.00	−0.1156	0.89	−11%
Survival	0.0177	0.0000	74.90%	1.76/3.17	0.8582	2.36	136%

**Table 2 plants-13-01913-t002:** Heterogeneity statistics and fold change under non stress conditions. POD = Peroxidase activity; SOD = Superoxide dismutase activity; CAT = Catalase activity; APX = Ascorbate peroxidase activity; RWC = Relative water content. Q test = *p*-value of the Q statistic (which measures heterogeneity) under the null hypothesis that there is no true heterogeneity among the studies. PI = Prediction interval; $\ln\frac{R}{R}$ = Response Ratio effect size; exp = to make the data more intuitive we have exponentiated the effect size with the formula $\frac{R}{R}\;=\;e^{\ln\left(\frac{R}{R}\right)}$; % change = percentage fold change obtained with the formula: (exp − 1) × 100.

Parameter	$\tau^{2}$	Q Test	$I^{2}$	PI	$\ln\frac{R}{R}$	Exp	% Change
Proline	0.0218	0.0000	71.60%	1.25/2.29	0.5241	1.64	64%
Plant height	0.0027	0.0000	68.79%	0.92/1.16	0.0346	1.04	4%
Seed number	0.0063	0.0000	73.94%	0.88/1.25	0.0457	1.05	5%
Seed weight	0.0038	0.0343	32.26%	0.89/1.20	0.0348	1.04	4%
Chlorophyll	0.0012	0.0025	54.37%	0.95/1.11	0.0265	1.03	3%
Root length	0.0019	0.0000	0.68%	1.09/1.12	0.1032	1.11	11%
Plant weight	0.0005	0.0000	62.94%	0.96/1.07	0.0124	1.01	1%
POD activity	0.0000	0.6647	0.00%	0.99/1.15	0.0615	1.06	6%
SOD activity	0.0127	0.0000	83.16%	−0.24/0.24	0.0005	1.00	0%
MDA activity	0.0055	0.0000	80.07%	0.84/1.09	−0.0468	0.95	−5%
CAT activity	0.0396	0.0000	74.29%	−0.29/0.67	0.1416	1.15	15%
APX activity	0.0434	0.0000	84.32%	0.91/1.34	0.0980	1.10	10%
RWC	0.0000	0.9893	0.00%	0.97/1.01	−0.0060	0.99	−1%
Stomatal conductance	0.0032	0.0204	48.52%	0.91/1.20	0.0421	1.04	4%
Electric conductivity	0.0000	0.9583	0.00%	0.94/1.00	−0.0302	0.97	−3%
Survival	0.0000	1.0000	0.00%	0.94/1.06	0.0000	1.00	0%

## Data Availability

All the datasets used for this work are in the Appendix A.

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
