# Peer review of "Proline Metabolism Genes in Transgenic Plants: Meta-Analysis under Drought and Salt Stress"

_plants, 2024, doi:10.3390/plants13141913_

Round 1
Reviewer 1 Report
Comments and Suggestions for Authors
Proline is an important stress resistant substance produced by plants, and there is no more novelty in this article. The author only collected and summarized literature to propose their own ideas, without conducting any experimental verification of the reliability of the ideas. And the actual application value of this idea also needs further evaluation。So, I do not believe that this paper is acceptable.
Author Response
Reviewer 1)
Proline is an important stress resistant substance produced by plants, and there is no more novelty in this article. The author only collected and summarized literature to propose their own ideas, without conducting any experimental verification of the reliability of the ideas. And the actual application value of this idea also needs further evaluation。So, I do not believe that this paper is acceptable.
Reply to reviewer 1) Thank you for your frank (although negative) opinion and useful criticism. I am glad to know that you share the idea that proline is an important stress molecule produced by plants, but I beg to disagree with the lack of novelty of the manuscript. Despite the general belief that proline confers stress tolerance, all the evidence is correlative, and it is still controversial whether proline leads to stress tolerance or is merely a consequence of stress. Also unknown are the molecular mechanisms of proline-induced stress tolerance (if any), the complex relationships with ROS (proline removes ROS in the cytoplasm, but generates them in the mitochondrion), and the connections between stress response and plant development. In an effort to address these questions, we performed a meta-analysis, a statistical procedure that collects and summarizes literature data to integrate their results and quantitatively evaluate the effect of proline on drought and salinity tolerance in plants.
The goal of a meta-analysis is to objectively measure the "effect size" (direction and intensity of the pool of effects) to critically evaluate conflicting results, plan future research directions, and try to answer open questions. The aim of a meta-analysis is not to propose personal ideas, perform experimental verifications, or evaluate the feasibility of a research idea.
We believe, on the contrary, that within the limits of this type of analysis, our results have provided useful indications to clarify some controversial effects of proline both in normal and stress conditions, strongly reinforcing the idea, for example, that metabolism and not proline is involved in conferring tolerance to stress, confirming the importance of antioxidant enzymes, and reducing the importance of the compatible osmolyte function generally attributed to proline.
As to future research directions, the clear indication provided by this work is to consider possible manipulation of proline metabolic genes (likely with updated molecular techniques such as CRISPR/Cas9 gene editing) to improve drought and salt stress tolerance in crops. Because of the high heterogeneity, and significant effects of moderators found in this meta-analysis, however, a tailored, case-by-case approach for proline-related genetic modification is strongly recommended.
Reviewer 2 Report
Comments and Suggestions for Authors
I have thoroughly reviewed the manuscript. It is a well-described study with a nice set of data. However, in its present form, it cannot be published due to some limitations, which I have mentioned in the attached annotated file.
I would like to review the revised version to confirm that the changes have been made.

Author Response
REVIEWER 2
I have thoroughly reviewed the manuscript. It is a well-described study with a nice set of data. However, in its present form, it cannot be published due to some limitations, which I have mentioned in the attached annotated file.
I would like to review the revised version to confirm that the changes have been made.
Reply to reviewer 2) In addition to the revised version, we have attached an annotated version of the revised PDF to reply, point to point, to the annotated comments.
Comment 1) Add a general introduction about the problem, then focus on the topic as described below.“ Rapidly increasing global population and the challenges posed by anthropogenic and climate change, exert immense pressure on agriculture to meet the growing demand for food while also ensuring environmental sustainability. To address this, it is crucial to explore novel approaches in agriculture (Waqas et al., 2023), and understand plant mechanisms that can enhance crop yield sustainably. Environmental stressors, including drought, salinity, heat, and nutrient deprivation, trigger complex up- and downregulation of physiological and genomic pathways in plants. However, Proline accumulation upon environmental stress”
Reply 1) Thank you for your helpful comment. We have substituted the initial paragraph with a new one, which I have slightly rephrased from your suggestion to introduce better the general problem. We have also included two more references, including the one suggested by you.
Comment 2) Require additional references to bolster the discussion,
Also, discuss in one sentence that plants, like halophytes, mostly rely on inorganic ions for osmotic adjustment rather than organic ions (such as proline). Since organic ions synthesis requires energy for synthesis, as a result, plants face yield penalties (Waqas et al., 2021).
Waqas, M., Yaning, C., Iqbal, H., Shareef, M., Rehman, H. U., & Bilal, H. M. (2021). Synergistic consequences of salinity and potassium deficiency in quinoa: Linking with stomatal patterning, ionic relations and oxidative metabolism. Plant Physiology and Biochemistry, 159, 17-27. https://doi.org/10.1016/j.plaphy.2020.11.043
Reply 2) To add additional references and bolster the discussion, we have elongated the discussion, adding a large piece of text including more references and a sentence about the halophytes, as per your suggestion.
Comment 3)
after this , add this line , It is also crucial to explore the proline metabolism under combine stress conditions like drought and salinity (Hassan et al., 2023)
Reply 3) Thanks once more for your helpful input. We have included the suggested line. However, we made a change from (Hassan et al., 2023) to (Iqbal et al., 2023) for consistency in the reference style.
Comment 4) After this, add below line,
Furthermore, advanced approaches are highly warranted, which offer promising results. For instance, a spatiotemporal mapping of leaf apoplastic ion and metabolite patterns is a valuable approach that enables the simultaneous capture of numerous compounds, including ions, metabolites, proteins, hormones, and others, with a single sampling event. Consequently, it advances the study of the leaf apoplast’s roles in cell-cell communication, and conduit for metabolite trafficking like proline (Franzisky et al., 2023).
Reply 4) We have added the sentences. Thanks for your suggestion
Comment 5) Write your conclusion with a separate heading and include a clear take-home message
Reply 5) We have written a separate conclusion paragraph with a clear take-home message. Thank you for your suggestion.
Comment 6) It would be even better if you included another figure that illustrates the proline mechanism under stress conditions. This will attract the reader’s attention and contribute to a more citations of article.
Reply 6) We have added a new figure as per your suggestion. We have added a new figure as per your suggestion. For the same reason, we have given some color to Figure 4. Thanks for the advice.

Reviewer 3 Report
Comments and Suggestions for Authors
The amino acid proline accumulation in plants during abiotic stresses such as drought and salinity and is a reliable marker of environmental stress. While its accumulation is well established, its precise role in stress tolerance and its underlying molecular mechanism remain less clear. To address these issues, this study performed a meta-analysis - a robust statistical technique that synthesizes results from multiple independent studies while accounting for experimental differences. Their results suggest that the benefits of proline may be related to proline catabolism or may only become apparent during stress, possibly due to interactions with reactive oxygen species (ROS), which accumulate predominantly under stress conditions. In general, this manuscript is well written and produced a significant results, I have the following
Comments:
1. For the each word, the first letter should be capital;
2. There is no reference from 2024, literature need to be updated;
3. In the material part, who set all the measures are expressed as fresh weigh, normally, dry weight data are more reliable?
4. In the final discussion part, the authors should stress the importance or the significance of your study results.
Author Response
Comments and Suggestions for Authors
The amino acid proline accumulation in plants during abiotic stresses such as drought and salinity and is a reliable marker of environmental stress. While its accumulation is well established, its precise role in stress tolerance and its underlying molecular mechanism remain less clear. To address these issues, this study performed a meta-analysis - a robust statistical technique that synthesizes results from multiple independent studies while accounting for experimental differences. Their results suggest that the benefits of proline may be related to proline catabolism or may only become apparent during stress, possibly due to interactions with reactive oxygen species (ROS), which accumulate predominantly under stress conditions. In general, this manuscript is well written and produced a significant results, I have the following
Comments:
Comment 1. For the each word, the first letter should be capital;
Reply 1) Many thanks for spotting these mistakes. We have capitalized the first letter in all the figures and tables, both in the main text and in supplementary material, except letters that should be written in lowercase (e.g., "n" or "p-value").
- There is no reference from 2024, literature need to be updated;
Reply 2) We have added 23 more references in the manuscript, including four from 2023 and three from 2024.
- In the material part, who set all the measures are expressed as fresh weigh, normally, dry weight data are more reliable?
Reply 3) To the best of my knowledge, there are no major differences in reliability between dry weight and fresh weight, although in most cases people choose fresh weight because it is less labor intensive. Regarding meta-analyses, however, practically all analyze data expressed in fresh weight, because the greater number of fresh weight data guarantees greater statistical reliability to meta-analyses.
Round 2
Reviewer 1 Report
Comments and Suggestions for Authors
The author collected and summarized literature to propose their own ideas, without conducting any experimental verification of the reliability of the ideas. And the actual application value of this idea also needs further evaluation。The entire paper appears to focus more on the application of applied mathematics in botany. Additionally, I did not find the formula derivation process in the paper. I am uncertain if this paper is suitable for this journal. If the editorial department acknowledges the value of this paper, please seek further evaluation from experts in the relevant field.
Author Response
Reviewer 1:
The author collected and summarized literature to propose their own ideas, without conducting any experimental verification of the reliability of the ideas. And the actual application value of this idea also needs further evaluation。The entire paper appears to focus more on the application of applied mathematics in botany. Additionally, I did not find the formula derivation process in the paper. I am uncertain if this paper is suitable for this journal. If the editorial department acknowledges the value of this paper, please seek further evaluation from experts in the relevant field.
Reply to Reviewer 1:
Despite your negative criticism, we appreciate your sincere feedback and thank you for your suggestions.
However, we keep thinking that you have misunderstood the spirit of a meta-analysis, which is not intended to verify experimental hypotheses but aims to scrutinize the scientific literature to objectively evaluate the overall effects of multiple papers on the effects of proline on drought tolerance. By synthesizing existing research, we try to identify patterns and gaps in the literature, confirm or refute existing hypotheses, guide subsequent experimental work, and inspire targeted studies and practical applications.
On the contrary, we appreciated and followed your request to know the details of the formulas and algorithms and provided all R codes and database used in this meta-analysis as supplementary material.
Finally, regarding your criticism of the “too mathematical” nature of this meta-analysis, I must admit that you are right. Unfortunately, however, all meta-analyses have the same structure even when they focus on biological questions. So the question is whether or not a biological journal accepts meta-analyses.
Reviewer 2 Report
Comments and Suggestions for Authors
The authors have meticulously addressed all suggested changes, incorporating detailed revisions and enhancements that have substantially improved the clarity, quality, and overall impact of the manuscript, which is now deemed suitable for publication.
Author Response
Coomment 1:
The authors have meticulously addressed all suggested changes, incorporating detailed revisions and enhancements that have substantially improved the clarity, quality, and overall impact of the manuscript, which is now deemed suitable for publication.
Response to comment 1: Thank you for your positive considerations.
Round 3
Reviewer 1 Report
Comments and Suggestions for Authors
No comments